# A small molecule inhibitor of Rheb selectively targets mTORC1 signaling

Sarah J. Mahoney [1], Sridhar Narayan[1], Lisa Molz[1], Lauren A. Berstler[1], Seong A. Kang[1], George P. Vlasuk[1] & Eddine Saiah[1]

The small G-protein Rheb activates the mechanistic target of rapamycin complex 1 (mTORC1) in response to growth factor signals. mTORC1 is a master regulator of cellular growth and metabolism; aberrant mTORC1 signaling is associated with fibrotic, metabolic, and neurodegenerative diseases, cancers, and rare disorders. Point mutations in the Rheb switch II domain impair its ability to activate mTORC1. Here, we report the discovery of a small molecule (**NR1**) that binds Rheb in the switch II domain and selectively blocks mTORC1 signaling. **NR1** potently inhibits mTORC1 driven phosphorylation of ribosomal protein S6 kinase beta-1 (S6K1) but does not inhibit phosphorylation of AKT or ERK. In contrast to rapamycin, **NR1** does not cause inhibition of mTORC2 upon prolonged treatment. Furthermore, **NR1** potently and selectively inhibits mTORC1 in mouse kidney and muscle in vivo. The data presented herein suggest that pharmacological inhibition of Rheb is an effective approach for selective inhibition of mTORC1 with therapeutic potential.

[1] Navitor Pharmaceuticals, Inc., 1030 Massachusetts Ave. #410, Cambridge, MA 02138, USA. Sarah J. Mahoney and Sridhar Narayan contributed equally to this work. Correspondence and requests for materials should be addressed to S.J.M. (email: smahoney@navitorpharma.com)

The mechanistic target of rapamycin complex 1 (mTORC1) acts as a central regulator of metabolic pathways that drive cellular growth. mTORC1 carries out this function by sensing and integrating diverse inputs such as nutrients, growth factors, and energy status. The functional output of mTORC1 signaling impacts an array of cellular processes such as protein synthesis and degradation, lipid synthesis, nucleotide synthesis, cell proliferation, and autophagy[1,2]. mTORC1 is comprised of mTOR, the catalytic kinase subunit of the complex, and four additional subunits (Raptor, PRAS40, mLST8, and Deptor) that regulate its activity and access to substrates. mTOR is also present in a second, distinct complex termed mTORC2. The function of mTORC2 is less well understood, but is activated by growth factors and is important for cell survival, proliferation, and cytoskeleton organization[3].

Aberrant mTORC1 signaling has been implicated in the aging process[4] as well as several chronic diseases such as fibrotic disease (e.g., idiopathic pulmonary fibrosis[5]), metabolic disease (e.g., type 2 diabetes and obesity[6]), neurodegenerative disease (e.g., Huntington's and Alzheimer's Disease[7,8]), and autoimmune disorders (e.g., lupus)[9] as well as certain cancers[10] and rare diseases including tuberous sclerosis (TSC) and lymphangioleiomyomatosis (LAM)[11]. The allosteric mTORC1 inhibitor rapamycin and its various synthetic analogs (rapalogs) have been extensively used as clinical immunosuppressants, anti-cancer agents, and as a treatment for TSC and LAM[12]. However, the use of rapamycin/rapalogs at clinically approved doses have been reported to be associated with various adverse effects including hyperglycemia, hyperlipidemia, insulin resistance, wound healing, renal function, and hypertension among others[13]. While rapamycin/rapalogs are selective inhibitors of mTORC1 following acute treatment, it has been proposed that the downregulation of mTORC2 activity upon chronic treatment may be the basis for at least some of these adverse clinical effects including insulin resistance[14,15]. In support of this hypothesis, direct mTOR kinase inhibitors that equally inhibit both mTORC1 and mTORC2 show a similar adverse effect profile[16–19]. Therefore, there is a need for a selective inhibitor of mTORC1 which can be used more broadly in the treatment of chronic diseases without the associated adverse events due to the downregulation or direct inhibition of mTORC2.

Growth factor signaling into mTORC1 is regulated by the heterotrimeric TSC complex, which acts as a GTPase-activating protein (GAP) for Ras homolog enriched in brain (Rheb), a GTP-binding protein that is broadly expressed in human and mammalian tissues[20]. A recent structural determination has elucidated a potential mechanism for Rheb activation of mTORC1. A cryo-EM structure of Rheb bound to mTORC1 revealed that the Switch II region of Rheb interacts with the mTOR N-heat, M-heat, and FAT domains. Conformational changes in these domains upon Rheb binding suggest an allosteric mechanism for activating mTORC1[21]. Given its selective and important role in mTORC1 signaling, we hypothesized that Rheb would be a suitable molecular target for the development of selective small molecule inhibitors of mTORC1. Modulation of Rheb is expected to have no direct impact on the mTORC2 complex.

Rheb is a monomeric protein with a molecular weight of about 21 kDa. As in the closely related small GTPase Ras, the flexible switch I region of Rheb undergoes a conformational change during nucleotide hydrolysis and exchange, while the switch II region remains largely unchanged[22]. Point mutations in the switch II region of Rheb (Y67A/I69A and I76A/D77A) impair its ability to activate mTORC1[23], implying that the switch II region is key to Rheb's function in mTORC1 activation.

Rheb also requires post-translational farnesylation at the C-terminus to correctly associate with the lysosomal membrane[24]. Interestingly, several farnesyltransferase inhibitors (FTIs), originally designed to block farnesylation of mutant Ras, have since been shown to inhibit Rheb prenylation, which may contribute to the anti-proliferative mechanism of these compounds[25,26].

However, the lack of selectivity of FTIs, which act on several different GTPases, limits their utility as specific Rheb-targeted agents[27,28]. To our knowledge, no small molecules that either directly bind Rheb or specifically modulate Rheb activity have been reported to date. Herein, we report the identification of **NR1**, a small molecule that directly binds Rheb in the switch II domain and selectively inhibits the activation of mTORC1. Furthermore, we show that its mechanism of action, distinct from that of rapamycin, may confer a therapeutic advantage.

## Results

**Identification of Rheb-binding small molecules**. At the outset of these studies, no information was available concerning potential binding sites on Rheb outside of that for the guanine nucleotide. Among the large family of small GTPases[29], Ras has been the subject of intense research due to its frequent upregulation in many human cancers[30]. Direct-binding inhibitors of Ras[31] and other small GTPases such as Ral[32] and Rho[33] that have functional activity have been reported. Another common approach is to target the corresponding guanine-nucleotide exchange factor (GEF) or its interaction with the GTPase[34]. However, no GEF for Rheb has been identified to date[20]. Moreover, no structural information is available on other components of the mTOR pathway implicated in interactions with Rheb such as the TSC complex. Therefore, we chose to focus our efforts on discovering a direct-binding inhibitor of Rheb.

In the past two decades, fragment-based lead generation (FBLG) has emerged as a powerful method to rapidly identify small molecule binders of proteins of interest, and we chose this approach to identify small molecule Rheb inhibitors[35]. A nuclear magnetic resonance (NMR)-based saturation transfer difference (STD) assay was used to screen a library of 1600 diverse compounds. The STD method relies on the transfer of saturation from the protein to transiently-bound ligands via spin diffusion through an intermolecular nuclear Overhauser effect[36]. Compounds that showed greater than 10% difference in area between the saturation spectrum and the reference spectra (STD factor) of the largest peaks were considered binders (Fig. 1a).

Heteronuclear multi-quantum coherence (HMQC) spectroscopy was then utilized to validate hits and to determine the likely binding site(s) on Rheb. Two-dimensional (2D) $^1$H$^{15}$N HMQC spectra of isotopically labeled $^{15}$N-Rheb were measured in the presence or absence of fragment hits. Of the 20 STD NMR hits characterized in this manner, 7 showed significant perturbations in the chemical shifts of multiple residues. In particular, **1** (Table 1) showed significant chemical shift perturbations (>0.03 ppm) of 25 residues, with the most significant being Thr-61, Gly-63, Gln-64, Asp-65, Ser-75, Ile-76, Ile-78, and Val-107, all of which are present in or near the switch II domain (Fig. 1b). Figure 1c highlights these residues in the NMR structure of Rheb reported by Karassek et al.[37].

Further confirmation of the binding site of **1** was obtained through X-ray crystallography. Crystal structures of GDP- and GTP-bound Rheb have been reported, and the switch II domain is typically disordered and adopts different conformations. We obtained a 2.05 Å structure of GDP-bound Rheb (Supplementary Fig. 1) and replicated the results reported by Yu et al.[22]. **1** was then soaked into the same crystals, whereupon additional electron density was observed in the switch II domain (Supplementary Fig. 2). The 1.56 Å crystal structure of Rheb bound to **1** (Fig. 1d)

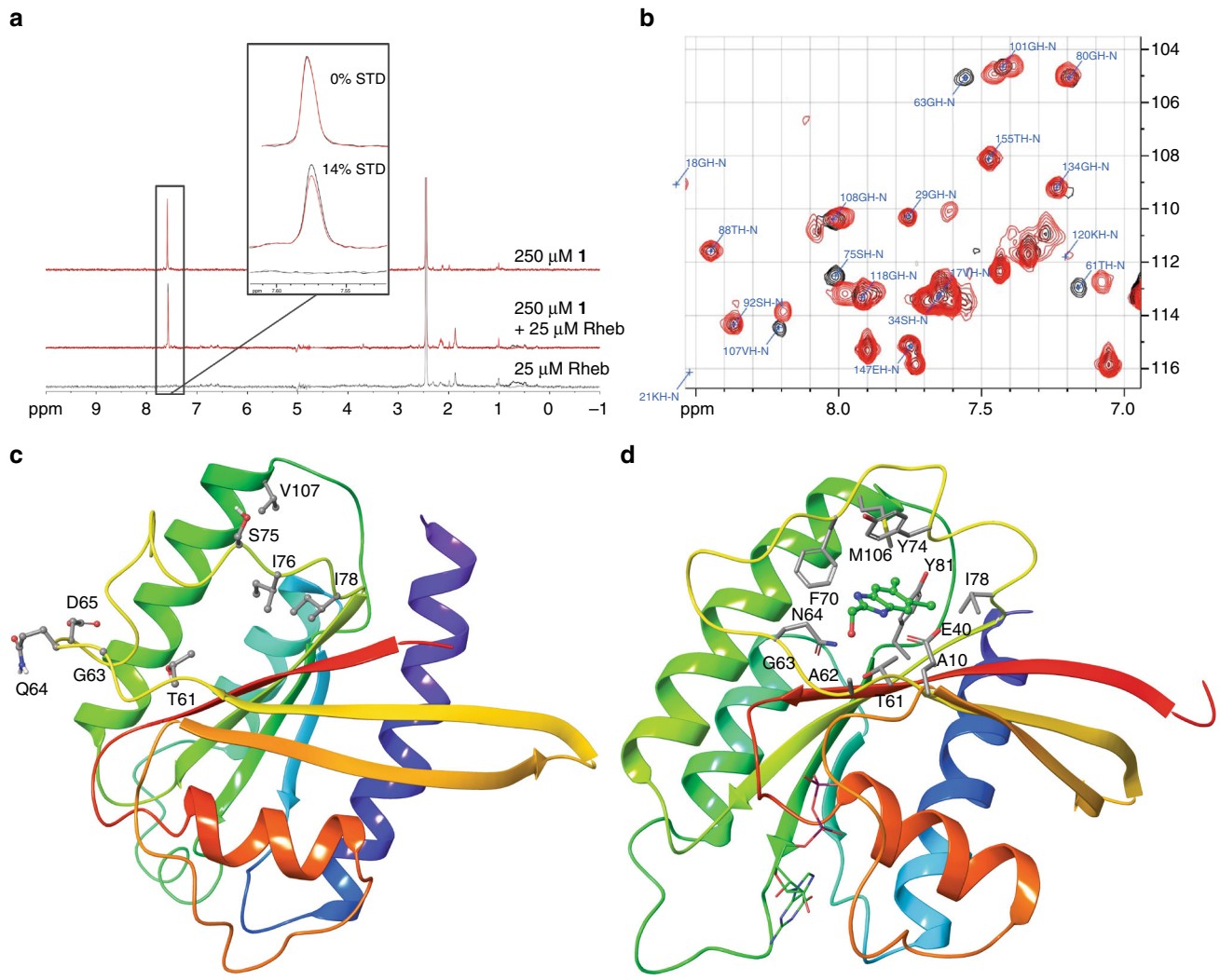

**Fig. 1** Fragment-based lead generation of Rheb-binding small molecules. **a** 1D [1]H saturation transfer difference (STD) NMR spectra of **1** in presence and absence of C-terminal truncated Rheb. **b** 2D [1]H–[15]N heteronuclear multi-quantum coherence (HMQC) NMR spectra of [15]N-labeled Rheb (200 μM) alone (black) and in presence of 1 mM **1** (red) showing key residues of the switch II domain. **c** Structure of Rheb showing side-chains of residues where significant chemical shift perturbations were observed upon binding **1**. NMR structure of GDP-bound Rheb from Karassek et al.[37] shown. **d** 1.56 Å X-ray crystal structure of GDP-Rheb with **1** bound in the switch II domain. Side-chains of residues within 4 Å of **1** are shown

shows a significant ordering of the switch II loop, forming 1 turn of a helix. The dichlorophenyl moiety anchors **1** into the hydrophobic pocket of the switch II domain, while the benzimidazole nitrogens form weak interactions with the carboxylate of Glu-40 and the backbone carbonyl of Ile-69. In addition, the primary alcohol interacts with the backbone N–H of Ala-62 via a bridging water molecule. The FBLG efforts described above provided us with a small molecule Rheb binder and confirmed that the binding site was in the switch II domain. Our observations were in line with those of Maurer et al.[38], who reported small molecule inhibitors that bind in a similar region of KRas. Given the importance of this domain in mTORC1 signaling[23], we were encouraged about the prospects of using **1** as a starting point for our drug discovery efforts. However, due to their weak binding affinity, we did not expect **1** or its close analogs to have functional activity. Therefore, hit to lead optimization was carried out to develop a more potent Rheb binder.

**Hit to lead optimization**. We first sought to design analogs of **1** with improved binding affinity towards Rheb using a structure-

based drug design (SBDD) strategy. An analysis of the X-ray structure of **1** bound to Rheb showed several areas for modification with potential to gain additional interactions. We focused on installing substitutions at the C2 and N1 positions to gain additional H-bonding or polar interactions and on installing hydrophobic groups on the phenyl ring to optimally fill the hydrophobic cavity of the binding site.

We used both Rheb binding and functional activity to guide the SAR optimization. Due to its high sensitivity and ability to detect weak binding ligands, STD NMR was initially used to measure Rheb binding. In addition to the benzimidazole of **1**, we also explored the related indole scaffold, which allowed a wider range of functionalization. This allowed us to design compounds **2**, **3**, and **4** (Table 1) which showed improved affinity towards Rheb. As the medicinal chemistry program progressed, STD NMR was found to be less reliable for more lipophilic compounds with lower solubility. Therefore, we also used affinity selection mass spectrometry (ASMS) as a semi-quantitative measure of Rheb binding[39].

To measure functional activity, we developed an in vitro assay for Rheb-dependent mTORC1 kinase activity (Rheb-IVK). To

**Table 1 Structure–activity relationships of select compounds showing Rheb binding and functional activity**

| Structure | Compound | %STD GDP[a] | %STD GMPPNP[b] | ASMS %R$_3$/R$_0$[c] | IVK IC$_{50}$ (µM)[d] |
|---|---|---|---|---|---|
| | **1** | 21 | 13 | 1.9 | > 500 |
| | **2** | 24 | 13 | 1.9 | 350 |
| | **3** | 49 | 32 | 7 | 84 |
| | **4** | 57 | 48 | 21 | 30 |
| | **NR1** | ND | ND | 37 | 2.1 |

ND not determined
[a]STD NMR factor (%) for compound binding to Rheb loaded with GDP
[b]STD NMR factor (%) for compound binding to Rheb loaded with 5′-guanylyl imidodiphosphate (GMPPNP), a non-hydrolyzable analog of GTP
[c]Affinity selection mass spectrometry enrichment factor after three cycles
[d]IC$_{50}$ in the Rheb-IVK assay

accomplish this, we adapted a previously reported kinase assay[40] to a robust and high throughput format (Fig. 2a). This assay reconstitutes Rheb activation of mTORC1 in vitro using full-length Rheb and mTORC1 purified from mammalian cells via a Flag-Raptor handle. mTORC1 kinase activity was monitored by measuring $^{T37/T46}$p4E-BP1, a well-characterized mTORC1 substrate, using a specific antibody and Western blot analysis.

As shown in Fig. 2b, Rheb activated mTORC1 in vitro, but only when charged with GTP and not GDP. As expected, positive control Torin-1, a dual mTORC1/mTORC2 inhibitor that targets the ATP-binding site of mTOR, inhibited mTORC1 activity in the Rheb-IVK (Fig. 2b). We were encouraged to see that improvement in Rheb-binding affinity was accompanied by a corresponding increase in functional activity (Table 1). Interestingly, previously described mutants in and around the Switch II region[23,41] did not activate mTORC1 in the Rheb-IVK, supporting the importance of this domain in mTORC1 activation (Supplementary Fig. 3).

We further adapted the Rheb-IVK to a LanthaScreen TR-FRET endpoint to increase its sensitivity and throughput. The assay was optimized for both inputs of Rheb and mTORC1 (Supplementary Fig. 4a, b). Results obtained from the two assay formats were in good agreement as shown by comparative data for **3** and Torin-1 (Fig. 2b–d), thereby enabling the use of the Rheb-IVK as a primary assay for compound optimization. As expected, rapamycin showed no inhibition in the Rheb-IVK (Fig. 2e) due to the absence of the accessory protein FKBP12.

Our medicinal chemistry efforts culminated in the identification of **NR1** (Supplementary Figs. 5 and 6), which showed an IC$_{50}$ of 2.1 µM in the Rheb-IVK (Fig. 2f), along with significant affinity

for Rheb binding as evidenced by ASMS. Thus, an increase in binding affinity with concomitant functional activity was achieved by using the SBDD design principles outlined above.

To rule out direct inhibition of mTOR kinase activity independent of Rheb, we employed HotSpot, a standard radiometric assay that directly measures mTOR kinase catalytic activity toward substrate 4E-BP1[42]. **NR1** had no inhibitory activity in HotSpot up to a concentration of 30 µM, demonstrating its specificity for Rheb. By contrast, Torin-1 inhibited mTOR activity at an IC$_{50}$ of 21 nM (Supplementary Fig. 4c, d).

**Biophysical characterization of NR1**. We next sought to confirm binding of **NR1** to Rheb by X-ray crystallography. This was challenging due to the low aqueous solubility of the compound and its propensity to form soluble aggregates under many of the crystallization or soaking conditions. Using dynamic light scattering (DLS) to optimize conditions for soaking, it was found that β-octyl glucoside (BOG) was essential to prevent aggregation of **NR1** in solution. Single crystals of Rheb were soaked with **NR1** in the presence of BOG, and the crystals subsequently analyzed by X-ray diffraction. Compared to unbound Rheb, additional electron density was observed at the switch II domain in three of four Rheb molecules in the asymmetric unit (Supplementary Fig. 7). The final model was refined at a resolution of 2.6 Å (Fig. 3a).

An analysis of the X-ray crystal structure shows that the switch II loop opens significantly to accommodate the large **NR1** molecule. The 4-bromoindole core of **NR1** occupies the largely hydrophobic switch II pocket, making van der Waals interactions with the side-chains of Tyr-74, Pro-71, Phe-70, and Leu-12. Furthermore, the C2 carboxylate forms an unusual H-bond

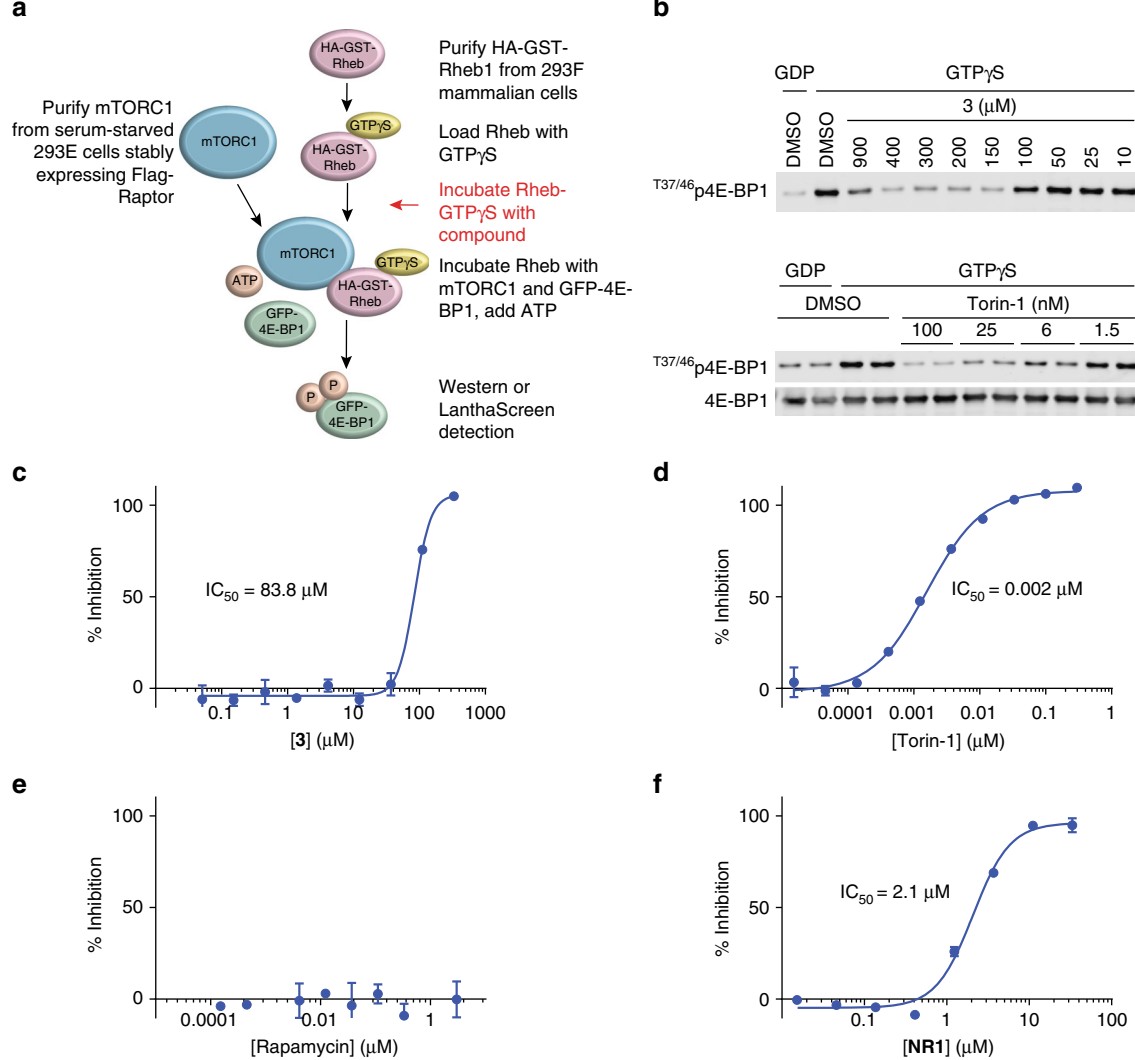

**Fig. 2** Rheb-binding compounds inhibit mTORC1 activity in vitro. **a** Schematic of Rheb-IVK (described in Methods section). **b** Evaluation of **3** and Torin-1 in the Rheb-IVK assay using Western blot analysis. Rheb-IVK reaction mixtures were incubated with either **3** or Torin-1 at the indicated doses, run on Western blots, and probed with T37/T46p4E-BP1-specific antibody. **c–f** Evaluation of **3**, Torin-1, rapamycin, and **NR1** in Rheb-IVK. The assay was run similar to the Western blot but processed in a LanthaScreen format using a Terbium labeled T46p4E-BP1 antibody. Error bars represent standard deviation from the mean of technical duplicates; graphs represent data from one of at least three separate experiments

interaction with the side-chain of Glu-40 (Fig. 3b). Other portions of **NR1** such as the dimethyl benzamide group and the dichlorothiophenyl substituent are solvent exposed in the X-ray structure, suggesting no significant interactions with Rheb.

Using a selection of Rheb mutants, Long et al. have identified key residues in the switch II domain involved in binding and activation of mTORC1[23]. Figure 3c shows the close interaction of bound **NR1** with these residues responsible for Rheb-dependent mTORC1 activation. It is therefore conceivable that portions of **NR1** positioned outside the Rheb-binding pocket sterically block the interaction with mTORC1. Moreover, conformational changes in the switch II loop upon **NR1** binding also likely make the interaction with mTORC1 energetically unfavorable. However, we have been unable to experimentally verify whether **NR1** acts by blocking the interaction of Rheb and mTOR due to the well-known technical challenges in detecting this interaction in a definitive manner. Figure 3d shows a stereo image of the electron density map at the binding pocket.

Despite low aqueous solubility, we carried out isothermal calorimetric (ITC) measurements to quantify the binding of **NR1**

to Rheb. **NR1** binds Rheb with a $K_D = 1.5\,\mu M$ and a 1:1 stoichiometry (Supplementary Fig. 8). A low heat of binding was observed for **NR1**, indicating that the binding energy was primarily entropically driven ($\Delta H = -0.315\,\text{kcal mol}^{-1}$; $T\Delta S = -0.7628\,\text{kcal mol}^{-1}$). This observation suggests that the binding of **NR1** to Rheb is primarily due to hydrophobic interactions, which is consistent with the intermolecular contacts observed in the X-ray crystal structure.

**NR1 selectively inhibits mTORC1 in cells**. We next evaluated the activity of **NR1** in cell culture. Selective inhibition of mTORC1 in cells produces a specific signaling signature characterized by a reduction in phosphorylation of the mTORC1 substrates S6K1 and 4E-BP1 and increase in phosphorylation of AKT, catalyzed by mTORC2. This distinctive profile reflects the relief of a feedback loop where active S6K1 directly phosphorylates IRS1, resulting in the degradation of IRS1 and subsequent reduction of growth factor signaling from RTKs to downstream effectors such as AKT[1]. Conversely, reduction in

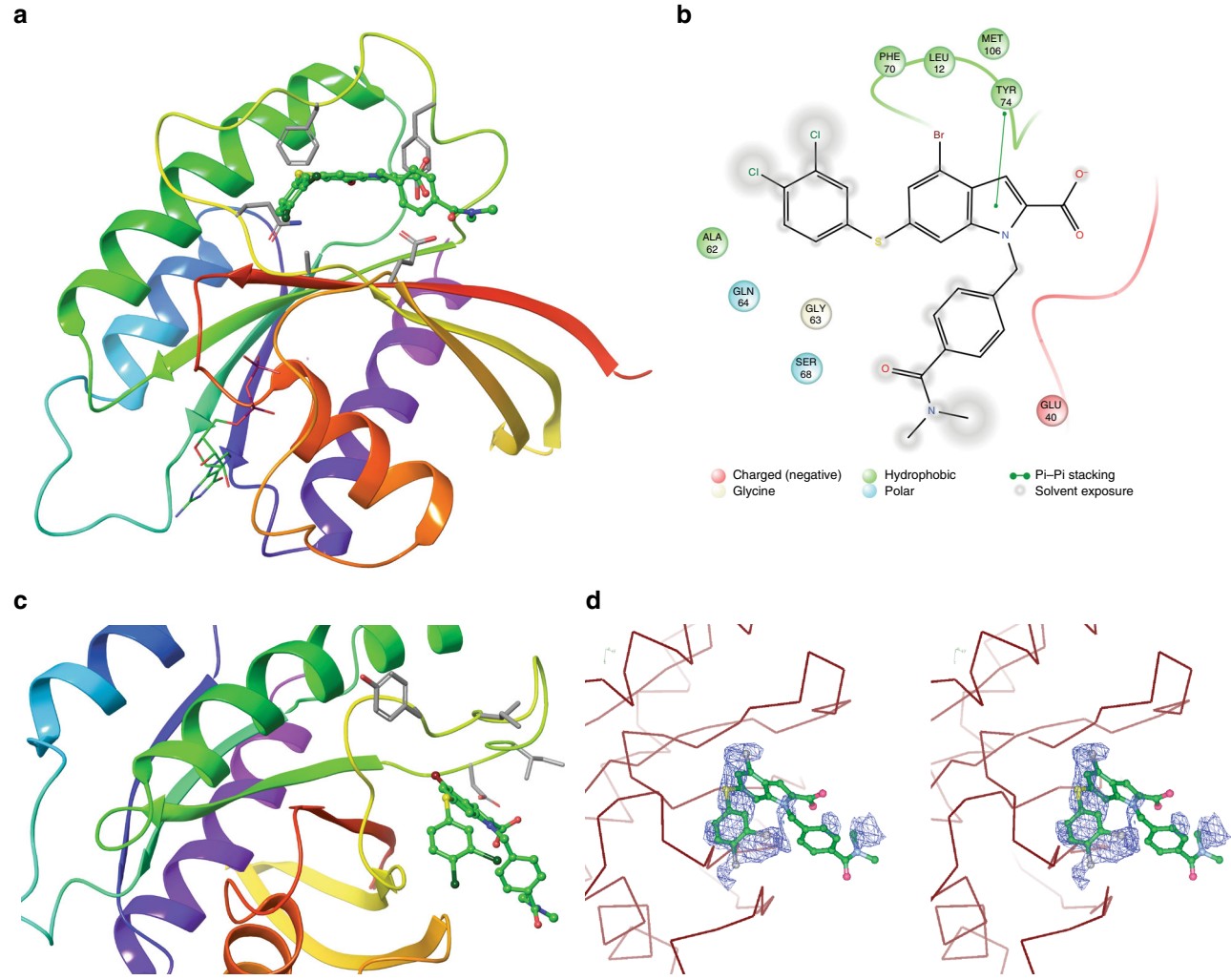

**Fig. 3** Biophysical characterization of **NR1**. **a** 2.6 Å X-ray crystal structure of C-terminal truncated Rheb bound to **NR1**. Side-chains of residues within 4 Å of **NR1** are shown in stick representation. **b** Ligand interaction diagram of **NR1** bound to Rheb; side-chains of residues within 4 Å of the ligand are shown. **c** Another view of the X-ray structure of **NR1** bound to Rheb; side-chains of residues implicated in activating mTORC1 are shown. **d** Stereo image of **NR1** bound to Rheb

$^{T389}$pS6K1 is expected to lead to an increase in $^{S473}$pAKT levels. We used AlphaLISA assays to detect $^{T389}$pS6K1 and $^{S473}$pAKT in MCF-7 cells. During the development of the assays, we noticed that $^{T389}$pS6K1 was more responsive to our compound series than $^{T37/46}$p4E-BP1, which may be due to differences in mTORC1 substrate classes in a cellular setting. This difference in sensitivity has previously been observed with indirect mTORC1 inhibition by nutrient and growth factor starvation and by rapamycin treatment[43,44]. Consistent with its activity in the Rheb-IVK, **NR1** inhibited the phosphorylation of $^{T389}$pS6K1 and increased the phosphorylation of $^{S473}$pAKT in a dose-dependent manner (Supplementary Fig. 9), suggesting that **NR1** is a selective inhibitor of mTORC1. These data were confirmed in a Western blot format with overnight serum-starvation and insulin stimulation (Fig. 4a). The Western blot assay also showed a downshift of the 4E-BP1 banding pattern, suggesting loss of phosphorylation. It is notable that the cellular potency of **NR1** is in the same range as that of its biochemical activity in the Rheb-IVK, strongly indicating a high degree of cell permeability of the compound.

A potential concern for the therapeutic use of Rheb inhibitors is their selectivity versus other small GTPases such as Ras. We used MCF-7 cells to evaluate the effect of **NR1** on the Ras/ERK pathway. Significantly, **NR1** did not inhibit EGF-induced phosphorylation of $^{T202/Y204}$pERK1/2, indicating that it does not inhibit Ras (Fig. 4b). Indeed, we observed an increase in $^{T202/Y204}$pERK1/2, similar to what has been seen previously with mTORC1 inhibition[45]. To further explore the specificity for Rheb vs. other GTPases, we performed an in vitro activity assay with the related small GTPase Rap1, in which we pre-loaded the lysates with either GDP or GTPγS and then treated the lysates with **NR1**. A subsequent pulldown of Rap1 using GST-RalGDS-RBD indicated that **NR1** did not affect the activity of Rap1 in vitro (Fig. 4c).

To test whether **NR1** maintains its activity in patient-derived cells and to isolate the mTORC1 pathway, we tested it in TRI102, an immortalized angiomyolipoma cell line derived from a LAM patient with a homozygous loss of function mutation in TSC2[46,47]. Due to loss of TSC function, these cells have hyperactivated mTORC1 that is insensitive to growth factor activation or withdrawal signals. **NR1** inhibits mTORC1 signaling in TRI102 cells in a dose-dependent manner under both replete and serum-starved conditions (Fig. 4d) indicating that **NR1** acts downstream of TSC2 to inhibit mTORC1 signaling, further supporting its mechanism of action as a direct Rheb inhibitor.

To further support Rheb as the target of **NR1**, we assessed whether NR1 could inhibit $^{T389}$pS6K1 in cells transfected with constitutively active, Rheb-independent double mutants of

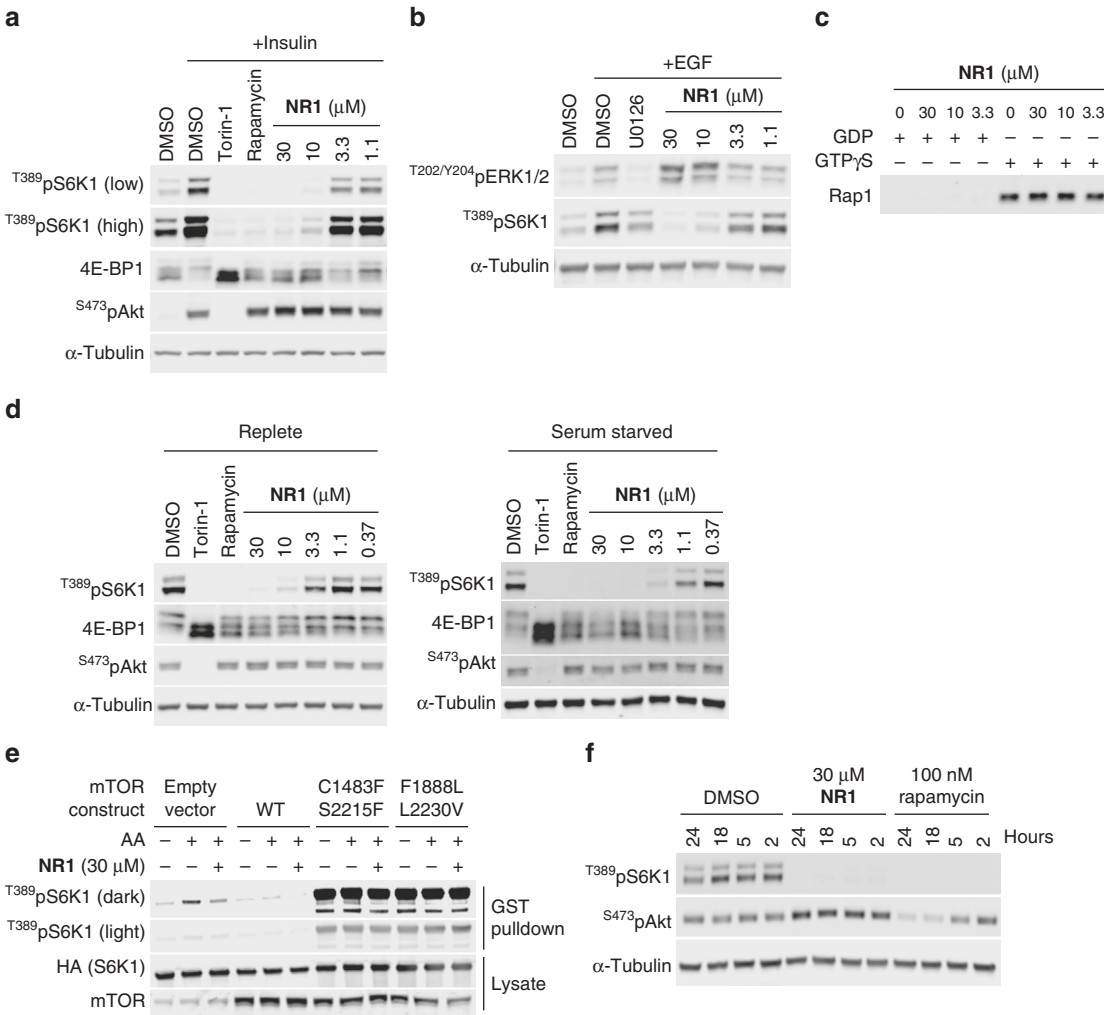

**Fig. 4 NR1** selectively inhibits mTORC1 but not mTORC2 in cells. **a**, **b** Effect of **NR1** on insulin-dependent activation of mTORC1 and EGF-dependent activation of $^{T202/Y204}$pERK1/2. MCF-7 cells were serum-starved for 16 h and treated with compound for 90 min before cell lysis and Western blot analysis. For insulin-stimulated cells, 100 nM insulin was added 30 min before cell lysis. For EGF-stimulated cells, 100 ng mL$^{-1}$ EGF was added 10 min before cell lysis. **c** Rap1 activity was detected in lysates loaded in vitro with either GDP or GTPγS and treated with **NR1** at the indicated concentrations. This was followed by a subsequent pulldown of GST-RalGDS-RBD. Bound Rap1 was detected by Western blot. **d** Evaluation of **NR1** in a LAM patient-derived cell line (TRI102) under replete and serum-starved conditions. For replete conditions, TRI102 cells were treated for 90 min with compound in DMEM + 10% FBS. For serum-starved conditions, TRI102 cells were serum-starved 16 h and then treated for 90 min with compound prior to cell lysis and Western blot analysis. **e** HEK293 cells co-transfected with the indicated mTOR constructs and HA-GST-S6K1 were starved of amino acids or treated with **NR1**. To assess mTORC1 activity, an HA IP was performed and Western blots probed for $^{T389}$pS6K1. **f** Chronic treatment of PC3 cells with **NR1** to evaluate the inhibition of mTORC1 (using $^{T389}$pS6K1) versus mTORC2 (using $^{S473}$pAKT). PC3 cells were treated with compound under replete conditions for 24 h prior to cell lysis and Western blot analysis. Rapamycin was used at 100 nM and Torin-1 at 250 nM

mTOR originally described by Xu et al.[48]. Figure 4e shows that, while **NR1** reduces $^{T389}$pS6K1 phosphorylation with vehicle or WT mTOR, it has no impact with the mTOR mutants. This result suggests that **NR1** acts upstream of mTOR to affect mTORC1 activity.

Long-term (>24 h) rapamycin treatment can lead to disruption or downregulation of mTORC2 signaling in some cell types[14], which, as discussed above, has been suggested as a cause for some of the adverse clinical effects of rapamycin. Accordingly, a hallmark of a selective mTORC1 inhibitor such as **NR1** would be the lack of mTORC2 inhibition irrespective of the duration of treatment. PC3 cells are sensitive to mTORC2 inhibition with long-term rapamycin treatment[14]. Indeed, upon treatment with rapamycin for 24 h, both $^{T389}$pS6K1 and $^{S473}$pAKT were lowered, indicating inhibitory effects on both mTORC1 and mTORC2, respectively. In contrast, there was no inhibition of $^{S473}$pAKT by

30 μM **NR1** observed over the same time course (Fig. 4f), which produced potent inhibition of $^{T389}$pS6K1. These data demonstrate that **NR1**, as a direct Rheb modulator, is a unique, selective small molecule inhibitor of mTORC1 activation and distinct from the other known classes of mTOR signaling pathway modulating compounds.

After verifying that **NR1** selectively inhibited pathway signaling consistent with an mTORC1 activation inhibitor, we then explored whether **NR1** could elicit functional activity consistent with this mechanism of action. Based on the dependence of cell size on mTORC1 activity[49], we assessed the effect of **NR1** on the size of Jurkat cells, a human T cell line grown in suspension and responsive to mTORC1 inhibition. After 48 h of incubation with the compound, **NR1** effectively reduced the size of Jurkat cells in a dose-dependent manner similar to the known mTOR pathway inhibitors Torin-1 and rapamycin (Fig. 5a).

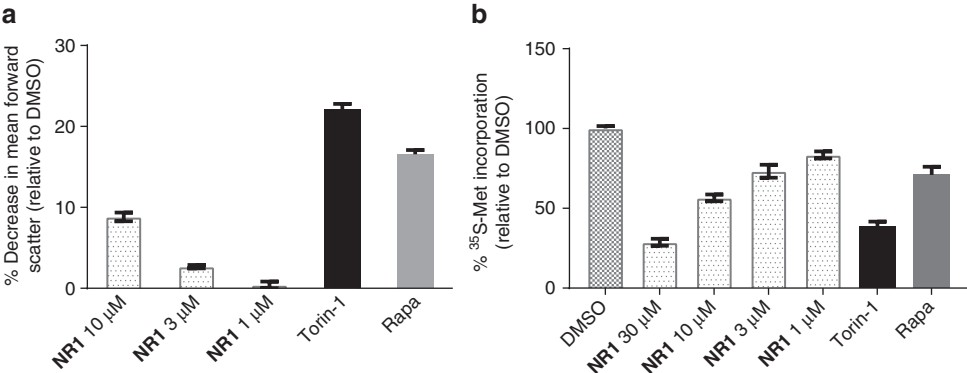

**Fig. 5** Functional outcome of mTORC1 inhibition by **NR1**. **a** Assessment of **NR1** effect on cell size in Jurkat cells compared to Torin-1 and rapamycin. Jurkat cells were treated with **NR1** over the indicated concentration range, Torin-1 (250 nM), or rapamycin (100 nM) for 48 h, and cell size was quantified as a measure of forward scatter using flow cytometry. Data are plotted as percent decrease in forward scatter. Error bars represent standard deviation of the mean of quadruplicates; graph represents data from one of three separate experiments. **b** Evaluation of **NR1** impact on protein synthesis compared to Torin-1 and rapamycin as measured by the incorporation of $^{35}$S-methionine into total protein. MCF-7 cells were treated with **NR1** at the indicated concentration range, Torin-1 (250 nM), or rapamycin (100 nM) for 2.5 h and labeled with $^{35}$S-methionine for 0.5 h. Error bars represent standard deviation of the mean of duplicates; graph represents data from one of two separate experiments

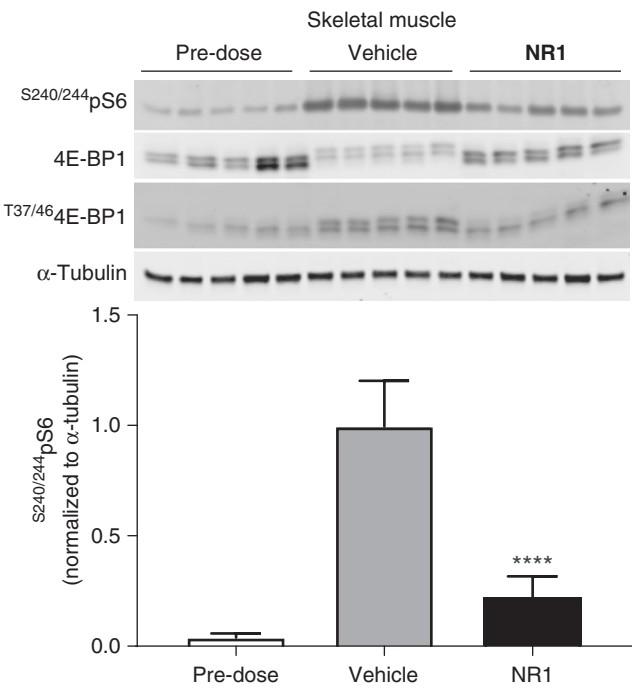

**Fig. 6** NR1 inhibits feeding-induced mTORC1 activation in skeletal muscle. Mice were starved for 16 h pre-dose to achieve a basal level of mTORC1 activity, then dosed with **NR1** (30 mg kg$^{-1}$ IP) or vehicle and allowed to re-feed ad libitum. After 2 h, gastrocnemius muscle was collected, and $^{S240/244}$pS6 levels were measured by Western blot and quantified. The results are shown as mean ± SEM ($n = 5$ per group). One way ANOVA for **NR1** with respect to vehicle are as follows: $F_{(2, 12)} = 73.8$, $p < 0.001$

Another functional correlate dependent on mTORC1 activity is protein synthesis[50]. To assess the impact of **NR1** compared to Torin-1 and rapamycin, we treated MCF-7 cells for 2.5 h with compounds and then labeled the cells with an $^{35}$S-Met labeling mix for 30 min. We observed that **NR1** could dose-dependently reduce protein synthesis in this assay to a similar extent as rapamycin and Torin-1 (Fig. 5b). The functional effects of **NR1** on cell size and protein synthesis are consistent with the observed selective inhibition of mTORC1 pathway signaling.

**NR1 inhibits the mTORC1 pathway in vivo**. The selective inhibition of mTORC1 activation by **NR1** in vitro and in cell-based systems prompted us to evaluate the effect of this compound on mTORC1 activity in vivo. Pharmacokinetic (PK) studies of **NR1** in mice indicated that both oral and intraperitoneal (IP) dosing could afford sustained drug levels that would be sufficient for inhibition of the mTORC1 pathway based on the observed cell-based activity. In particular, the plasma concentration of **NR1** was sustained over 5 μM for 2 h after IP administration of a 30 mg kg$^{-1}$ dose (Supplementary Fig. 10). We therefore carried out a pharmacodynamic (PD) study to evaluate the efficacy of **NR1** at inhibiting mTORC1 signaling in vivo. In this experiment, mice were starved overnight to achieve a low basal level of mTORC1 activation (pre-dose). After a 16 h fast, the animals were treated with either vehicle or 30 mg kg$^{-1}$ **NR1** and allowed to re-feed for 2 h before sacrifice and tissue collection. mTORC1 activity was measured by quantification of $^{S240/244}$pS6 by Western blot analysis with a specific antibody. We chose kidney and gastrocnemius muscle as target tissues due to the relevance of mTORC1 in renal cell carcinoma[51] and angiomyo-lipomas in TSC and LAM[12] and for its important role in the regulation of protein synthesis and muscle mass[52]. The results are summarized in Fig. 6 for skeletal muscle and Supplementary Fig. 11 for kidney. Compared to the vehicle-treated group, the **NR1**-treated animals had significantly reduced mTORC1 activity in both kidney and skeletal muscle, clearly indicating down-regulation of the pathway due to compound. In addition, there was a clear band shift for $^{T37/46}$4E-BP1 in skeletal muscle, further confirming the activity of **NR1** on the mTORC1 pathway.

## Discussion

The mTORC1 signaling complex is a central integrator of both growth factor and nutrient signals and directly regulates anabolic and catabolic activities in the cell, thereby maintaining metabolic homeostasis. Hyperactive mTORC1 that is no longer responsive to growth factor or nutrient signaling has been linked to multiple chronic diseases and directly contributes to the process of aging[4]. Thus, the therapeutic potential of mTORC1 inhibitors is widely acknowledged. However, to date, the only compounds targeting the mTOR signaling pathway that have found clinical utility have been in two classes: rapamycin and various analogs (rapalogs) or direct mTOR kinase inhibitors, both of which have adverse effects

resulting from the suppression of mTORC2 activity following chronic therapy.

Rapamycin and rapalogs bind to a cellular accessory protein FKBP12 to form a complex that allosterically inhibits mTORC1, resulting in the selective inhibition of certain downstream substrates[12,43]. While selective for mTORC1 following short-term exposure, this mechanism of inhibition also downregulates the formation of the mTORC2 complex in particular cell types and tissues[14]. The resultant reduction of mTORC2 activity has been linked to adverse effects observed clinically with chronic use of approved doses of rapamycin/rapalogs[15]. Likewise, the use of direct inhibitors of the mTOR kinase, which equally inhibit both mTORC1 and mTORC2, have also been associated with significant adverse effects when used clinically, limiting the broader use of this class of compounds beyond certain oncology indications[16–19].

A direct consequence of a reduction in mTORC2 activity with chronic rapamycin/rapalog use can be observed in the complex metabolic effects of these compounds in cells and in vivo. In a mouse model of type 2 diabetes and in acute treatment of diet-induced obese mice, rapamycin exacerbates insulin resistance[53,54]. Furthermore, long term use of rapamycin in patients and the resulting decreased glucose tolerance has been linked to increased insulin resistance[55]. Several groups have probed the requirement for mTORC1 and mTORC2 on glucose tolerance in key metabolic tissues and demonstrated that rapamycin-induced insulin resistance is independent of hepatic mTORC1 and mimicked by loss of mTORC2[15,56,57].

The abundance of data linking mTORC1 to many chronic diseases, combined with the apparent clinical risks of reducing the activity of mTORC2, clearly emphasizes the need to identify selective inhibitors of mTORC1 that do not disrupt mTORC2. To this end, there are several key aspects in the regulation of mTORC1 versus mTORC2 activity that suggests that this may be feasible. First, mTORC1 is uniquely responsive to the availability of nutrients such as amino acids and glucose. As such, specific protein sensors and regulatory complexes have been described[58–60]. In addition, activation of mTORC1 in response to growth factors is uniquely dependent on the GTP-binding protein Rheb[1]. Targeting either or both pathways has the potential to yield a novel class of selective inhibitors of mTORC1 activity, which may be significantly differentiated from the current class of mTOR pathway inhibitors.

To evaluate this strategy, we chose Rheb as the molecular target to selectively inhibit mTORC1. This was based in part on the availability of highly purified protein as well as both NMR[37] and X-ray crystal structures[22] of Rheb, which allows an SBDD approach. In addition, small molecule modulators of other small GTPases structurally related to Rheb, including Ras[38], Ral[32], and Rho[33], have been reported.

Using a FBLG approach and STD-NMR to screen for direct binders, we identified a small molecule hit compound (1). We also developed a high-throughput in vitro kinase assay (Rheb-IVK) to measure functional activity of Rheb binders. A very weak binder, 1, had no activity in Rheb-IVK. Nevertheless, we were encouraged by NMR studies that revealed 1 binding in the switch II domain, which contains residues previously implicated in activity of mTORC1[23]. Further optimization of the chemical series using both Rheb binding and functional activity as readouts led to the identification of NR1.

Several lines of evidence suggest that NR1 selectively engages Rheb in cells in the single-digit micromolar potency range. First, NR1 produces a distinctive signaling profile that is reminiscent of that seen in Rheb$^{-/-}$ knockout embryos and MEFs[61,62]. As seen with Rheb$^{-/-}$ cells, treatment of cells with NR1 selectively inhibits mTORC1 kinase, the direct target of Rheb, while activating

mTORC2 kinase (Fig. 4 and Supplementary Fig. 9). The activation of mTORC2, a related complex which shares the mTOR catalytic subunit with mTORC1, occurs through relief of an mTORC1-dependent negative feedback loop[1]. Second, NR1 is not a direct inhibitor of the mTOR kinase, as evidenced by the lack of inhibition of mTORC2 in cells and lack of direct inhibition of mTOR kinase in an in vitro kinase assay (Supplementary Fig. 4c, d). Additionally, we showed that NR1 could still inhibit mTORC1 signaling in TRI102 cells (Fig. 4d), suggesting that the target is in between TSC and mTORC1. Finally, NR1 does not inhibit Ras signaling in cells (Fig. 4b), even though Ras is a related small molecular weight G-protein, and it does not inhibit Rap1 activity in vitro (Fig. 4c).

NR1 is active in a variety of cell lines, including the patient-derived kidney cell line TRI102. Activity in this cell line not only shows potential patient relevance but also pinpoints compound activity downstream of the TSC complex since TSC2 is mutated in TRI102 cells. Moreover, NR1 showed no inhibition of the Ras/ERK pathway, indicating its selectivity for Rheb over Ras. Importantly, NR1 gave a unique signaling profile from rapamycin and ATP-competitive mTOR inhibitors—long-term treatment of cells with NR1 did not affect mTORC2 activity. This validates efforts to identify a selective inhibitor of mTORC1 activation and suggests that NR1 or related compounds could have a more favorable clinical profile compared with the current mTOR pathway drugs.

As with any new pharmacological agent, the anticipated side effects will need to be considered. The only data available from the literature that may help in understanding the potential side effects of reducing Rheb activity come from genetic knock-out studies which, because of their inherent impacts on developmental and other effects, cannot be directly related to what might be expected following pharmacological inhibition. Thus, careful development of a Rheb inhibitor would have to take into account the potential organ systems that may be impacted, such as neurological[63,64] and immunological[65,66].

NR1 also has reasonable drug-like properties that have allowed us to evaluate its efficacy in vivo. We found that this compound significantly reduced mTORC1 activity in skeletal muscle and the kidney. The observed reduction in mTORC1 activity in the kidney suggests this approach to selective inhibition of mTORC1 may be useful in TSC, LAM, sporadic forms of renal cell carcinoma in which TSC function is lost[67], and other forms of cancer that are driven by hyperactivated mTORC1. Furthermore, NR1 could also prove to be a useful research tool to further interrogate the biology of mTORC1 pathway regulation. In this regard, a constitutive transgenic model overexpressing TSC1 (TSC1$^{tg}$) resulting in a mild downregulation of mTORC1 activity in most tissues has recently been described[68]. The improved longevity and healthspan of these mice is directly relevant to the class of Rheb inhibitors described here based on the extent of mTORC1 inhibition observed in vivo (Fig. 6 and Supplementary Fig. 11).

Small GTPases have proven to be challenging drug targets[30]. Even in the case of Ras, which has been the subject of several decades' worth of efforts, no small molecule inhibitor has yet reached the clinic. Identification of direct-binding inhibitors of Ras has been particularly challenging due to the lack of deep hydrophobic pockets on the protein. Although many small molecule binders are known, a high-affinity ligand for Ras has yet to be identified. Ostrem et al. took advantage of a cysteine residue adjacent to the switch II domain to design covalent inhibitors of an oncogenic mutant KRas (G12C)[69]. More recently, Welsh et al. attempted to design multi-valent compounds that bind in shallow adjacent pockets of Ras[70]. Another interesting approach targeting Ral sought to take advantage of a binding pocket in the GDP-bound form of the protein[32]. In all cases, compounds with

micromolar affinity were discovered that showed functional activity.

Our results suggest that drugging Rheb is similarly challenging. Nonetheless, its key role in activating mTORC1 signaling makes Rheb a worthwhile therapeutic target. The present studies have resulted in the identification of a direct binder of Rheb and demonstrate the validity of targeting Rheb pharmacologically. We have shown that inhibiting Rheb directly results in a signaling profile distinct from current mTOR pathway inhibitors with respect to the inhibition of mTORC2, which in turn may translate into differentiated clinical benefits. More importantly, our studies also confirm the key role of Rheb in mTORC1 activation and suggest that the discovery of small molecules that target the Rheb-mTOR protein-protein interaction would be a fruitful avenue to pursue.

## Methods

**Protein expression and purification.** For the STD-NMR fragment-based screen, HMQC-NMR, ITC, and crystallography, human Rheb protein (1–169) was purified from BL21 (DE3) *Escherichia coli* using a His-Smt-Rheb expression vector. The cell paste was lysed in lysis buffer (50 mM HEPES pH 7.5, 0.5 M NaCl, 5% Glycerol, 1% CHAPS, EDTA-free protease inhibitor tablets, lysozyme to 1 mg mL$^{-1}$, benzonase 250 U), sonicated, and purified on a HiTrap Ni Chelating Column (GE Healthcare) using the following buffer: 50 mM HEPES pH 7.5, 0.5 M NaCl, 5% Glycerol, and imidazole increasing in concentration from 5 mM to 500 mM. The His-Smt tag was cleaved using a Ulp1 enzyme and dialyzed into 50 mM HEPES pH 7.5, 0.5 M NaCl, 5% glycerol. The resultant protein was again purified on a Ni Chelating column to remove the Ulp1 enzyme and His-Smt tag and then purified by size exclusion chromatography on a Sephacryl S-100 26/60 column into the following final buffer: 20 mM NaPhos pH 7.0, 100 mM NaCl, 5 mM MgCl$_2$, 2 mM TCEP. For HMQC NMR, $^{15}$N-labeled Rheb was generated using the same protocol but grown in minimal medium supplemented with $^{15}$N-labeled amino acids.

**Fragment screening and hit confirmation by NMR.** STD experiments were performed at 10 °C in pools of up to 10 fragments each at a concentration of 250 μM. Fragment pools were incubated with 25 μM Rheb in d$^6$-DMSO and D$_2$O in the following buffer: 20 mM NaHPO$_2$, 100 mM NaCl, 1 mM MgCl$_2$ at pH 7.0. Off- and on-resonance irradiation frequencies were set to 1 ppm and 30 ppm, respectively. The irradiation power of the selective pulses was 500 Hz, the saturation time was 3 s, and the total relaxation delay was 4 s; 16 scans were recorded and a total of 16,000 points were collected. The STD factor was calculated as STD (%) = 100 ($I_{ref} - I_{sat}$)/ $I_{sat}$, where $I_{ref}$ and $I_{sat}$ are integral areas of the reference and saturation peaks, respectively. Compounds with STD factor >10% of the largest peak were considered hits.

2D $^1$H $^{15}$N HMQC-SOFAST experiments were performed at 25 °C on samples containing 200 μM $^{15}$N-labled Rheb and 1 mM ligand in d$^6$-DMSO in the same buffer used for STD NMR. The acquisition parameters were as follows: 0.3 s recycle delay, 8 scans, 10,000 points, and 98 increments per scan. Combined chemical shift perturbations were calculated as $\Delta\delta = [\Delta\delta(^1H)^2 + (\Delta\delta(^{15}N)/5)^2]^{1/2}$.

**X-ray crystallography.** Rheb was crystallized by vapor diffusion in a solution of 22% PEG 3350 and 0.1 M sodium acetate at pH 5.5. In order to form a complex with **NR1**, a single crystal of Rheb was soaked for 10 h at 12 °C in a 4 μL drop composed of 27% PEG 1500, 0.1 M HEPES pH 7.0, 10 mM MgCl$_2$, 10 mM β-octyl glucoside (BOG) with soluble aggregates of **NR1** in the crystallization solution, as monitored by DLS. For X-ray data collection, a crystal was flash frozen in liquid nitrogen. Diffraction data were collected to a resolution of 2.3 Å at the Canadian Light Source (CLS) beamline 08B1 at 1.000 Å and 100 K. Data were processed and scaled with XDS and XSCALE. The structure was solved by molecular replacement with the program MOLREP and the coordinates of PDB code 1XTQ as a search model. The structure model was built using the program COOT and refined using the program Refmac. All data collection and refinement statistics are summarized in Supplementary Table 1.

**Software.** Crystallographic images were generated using Maestro (Schrödinger Release 2017-1), Schrödinger, LLC.

**NR1.** 4-bromo-6-(3,4-dichlorophenylthio)-1-(4-(dimethylcarbamoyl)benzyl)-1*H*-indole-2-carboxylic acid. To a solution of methyl 4-bromo-6-(3,4-dichlorophenylthio)-1-(4-(dimethylcarbamoyl)benzyl)-1*H*-indole-2-carboxylate (130 mg, 0.22 mmol) in THF/H$_2$O (4 mL: 4 mL), LiOH•H$_2$O was added (18.5 mg, 0.44 mmol), and then the mixture was stirred at room temperature for 17 h. The reaction was quenched with ice-water, the pH adjusted to 5 with 1 N HCl aqueous solution, and the resulting solution extracted with EtOAc, washed with brine, and dried over sodium sulfate. The combined extracts were concentrated and purified

by prep-HPLC to afford 4-bromo-6-(3,4-dichlorophenylthio)-1-(4-(dimethylcarbamoyl)benzyl)-1*H*-indole-2-carboxylic acid, **NR1** (118 mg, 92.9%), as a white solid. MS (EI+, *m/z*): 576.8 [M-H]$^+$. $^1$H NMR (500 MHz, DMSO) δ 13.49 (s, 1H), 7.86 (s, 1H), 7.60–7.41 (m, 3H), 7.30 (d, *J* = 8.1 Hz, 2H), 7.23 (s, 1H), 7.12 (dd, *J* = 8.5, 2.1 Hz, 1H), 7.03 (d, *J* = 8.1 Hz, 2H), 5.94 (s, 2H), 2.89 (d, *J* = 54.0 Hz, 6H). $^{13}$C NMR (125 MHz, DMSO) δ 107.1, 115.5, 116.3, 126.2, 126.8, 126.9, 127.1, 128.3, 129.1, 129.5, 131.2, 131.8, 135.1, 138.2, 139.7, 162.9, 169.8. HRMS (*m/z*): [M-H]$^-$ calculated for C$_{25}$H$_{18}$BrCl$_2$N$_2$O$_3$S, 574.9599; found 574.9633.

**Rheb-dependent mTORC1 in vitro kinase assay.** To purify mTORC1, HEK-293E cells (ATCC #CRL-10852) stably expressing FLAG-Raptor were serum-starved in DMEM (no FBS) overnight. The next day, the cells were collected and lysed in the following lysis buffer: 0.4% CHAPS, 150 mM NaCl, 50 mM HEPES pH 7.4, protease inhibitor (Sigma #11873580001). Lysates were cleared by centrifugation at 4 °C and incubated with anti-FLAG M2 affinity agarose gel (Sigma #A2220) for 1 h on nutator at 4 °C. The beads were washed three times in wash buffer (0.1% CHAPS, 120 mM NaCl, 50 mM HEPES pH 7.4), and elution was performed in mTORC1 elution buffer (150 mM NaCl, 50 mM HEPES pH 7.4, 0.03% CHAPS, 375 μg mL$^{-1}$ FLAG peptide). The eluate was separated from the beads using a Micro Bio-Spin column (BioRad #732-6204). The purified mTORC1 was stored at 4 °C for up to 1 week.

Human HA-GST-Rheb was purified from mammalian cells by first transfecting Freestyle 293-F suspension cells (ThermoFisher Scientific #R79007) grown in Freestyle 293 Expression Medium (ThermoFisher Scientific #12338018) with pRK5-HA-GST-Rheb DNA using polyethylenimine (Polysciences, 23966-2). Five days after transfection, the cells were lysed (150 mM NaCl, 50 mM HEPES pH 7.4, 1% Triton X-100, 5 mM MgCl$_2$, protease inhibitor). Lysates were cleared and incubated with glutathione agarose (Pierce #16100) for 1 h at 4 °C. HA-GST-Rheb was eluted from the glutathione agarose with elution buffer (150 mM NaCl, 50 mM Tris pH 8.0, 10 mM glutathione, 0.1% β-mercaptoethanol, 5 mM MgCl$_2$), concentrated, and stored at −80 °C.

For the Rheb-dependent in vitro kinase assay (Rheb-IVK), HA-GST-Rheb was first loaded with GTP-γ-S (Millipore #20–176) by incubating 2 μM HA-GST-Rheb with 10 mM EDTA and 0.4 mM GTP-γ-S at 30 °C for 10 min. MgCl$_2$ was added to a final concentration of 20 mM to stop the reaction. Loaded HA-GST-Rheb was then incubated with compound for 30 min at room temperature. mTORC1, GFP-4E-BP1 substrate (Life Technologies # PV4759), and kinase assay buffer (HEPES pH 7.4, KCl, MgCl$_2$) were added to the Rheb-compound mixture and incubated at room temperature for 20 min. ATP was then added, and the kinase reaction proceeded at 30 °C for 45 min before being stopped by the addition of SDS-PAGE loading buffer for Westerns and 10 mM EDTA for the LanthaScreen readout. The final concentrations of reagents are as follows: 100 nM loaded HA-GST-Rheb, compound (variable), 0.5 mM EDTA, 10 mM MgCl$_2$, 25 mM HEPES pH 7.4, 50 mM KCl, mTORC1 (fixed volume—0.1 μL in 10 μL total volume), 0.4 μM GFP-4E-BP1 substrate, 0.5 mM ATP.

For Western analysis of the kinase assay samples, blots were probed with a $^{T37/46}$p4E-BP1 antibody (Cell Signaling Technology #2855). For LanthaScreen (ThermoFisher #PV4757), the manufacturer's instructions were followed. Plates were read on an EnVision reader (PerkinElmer) using 495ex/520em filters (Life Technologies, #PV00315).

**Cell-based assays.** All cells used tested negative for mycoplasma. MCF7 (ATCC #HTB-22), TRI102, and PC3 (ATCC #CRL-1435) cells were grown in DMEM + 10% FBS. Jurkat Clone E6-1 cells were grown in RPMI1640 + 10% FBS. The antibodies used are as follows: $^{T389}$pS6K1 (Cell Signaling Technology #9234), $^{S473}$pAkt (Cell Signaling Technology #4060), 4E-BP1 (Cell Signaling Technology #9644), $^{T37/46}$p4E-BP1 (Cell Signaling Technology #2855), $^{T202/Y204}$pERK1/2 (Cell Signaling Technology #4370), tubulin (Sigma #T-5168), $^{S240/244}$pS6 (Cell Signaling Technology #5364). Torin-1 and rapamycin were obtained from LC Laboratories (#T-7887 and #R-5000, respectively), and U0126 was obtained from Sigma-Aldrich (#662005).

For Western-based signaling assays, cells were treated for either 90 min or 2–24 h as indicated before lysis, normalization, and Western analysis. Cells were lysed in Triton lysis buffer: 1% Triton X-100, 50 mM HEPES pH 7.4, 100 mM NaCl, 2 mM EDTA, 10 mM β-glycerophosphate, 10 mM Na-pyrophosphate, protease inhibitor.

For the cell size assay, Jurkat Clone E-61 cells were plated in 96-well plates concurrent with compound. The cells were incubated at 37 °C and 5% CO$_2$ for 48 h. PI stain (Invitrogen #727949) was added at a final concentration of 1 μg mL$^{-1}$, and cells were incubated for 15 min at room temperature. FACS analysis was performed; at least 10,000 events were collected for each sample. Viable cells (PI minus) were counted for FSC and SSC, and cell size (as measured by FSC) relative to the DMSO control was calculated.

For assessing mTORC1 activity with constitutively active mTOR mutants, HEK-293 cells (ATCC #CRL-1573) were transiently co-transfected with pRK5-HA-GST-S6K1 and FLAG-tagged WT or mutant mTOR constructs using the GeneJammer transfection reagent (Agilent #204132). Two days later, one set of cells were starved of all amino acids for 60 min in amino acid-free RPMI (MyBioSource #MBS653421) with 5 mM glucose and 10% dialyzed FBS (Life Technologies #26400044). Another set of cells were treated with 30 μM **NR1** for 90 min. After treatment, the cells were lysed in Triton X-100 lysis buffer, cleared, and

normalized. Immunoprecipitation of S6K1 was performed using glutathione agarose (ThermoFisher #16100). The IPs and lysates were run on Western blots which were then probed for [T389]pS6K1 (Cell Signaling Technologies #9234S), HA tag (Cell Signaling Technologies #3724), and mTOR (Cell Signaling Technologies #2972).

The protein synthesis assay was performed by incubation of MCF7 cells with compounds in MEM + 10% dialyzed FBS for 2.5 h and then incubated in labeling mix (compounds, Met- and Cys-free media, 10% dialyzed FBS, and [35]S protein labeling mix) for 30 min. Cells were lysed, proteins were precipitated, and resuspended pellets were read on a Microbeta 2.

For the Rap1 assay, HEK-293T cells (ATCC #CRL-3216) were serum-starved in DMEM (no FBS) overnight and lysed following the manufacturer's instructions (CST #8818). Lysate was loaded with nucleotide (1 mM GDP or 0.1 mM GTPγS) and then incubated with **NR1** for 30 min at room temperature. Precipitation of Rap1 and immunoblot were performed as directed.

Full images of Western blots are included in Supplementary Figs. 12–16.

**PK/PD studies**. This study was conducted in accordance with the requirements for the humane care and use of animals set forth in the Animal Welfare Act, the IACUC Guide for the Care and Use of Laboratory Animals, applicable Shanghai and state laws, and regulations and policies of Shanghai ChemPartner Ltd. The animals were not randomized, and the study was not blinded. Male C57BL/6 mice (5 per treatment group, 6–7 weeks old) were fasted of food overnight for 16 hours. IP injection of compound dissolved in MC Tween (0.5% methyl cellulose and 0.1% Tween-80 in water) was given, and then the mice were allowed to feed again ad libitum until sacrifice. The animals were euthanized with $CO_2$. Compound levels in the plasma of treated mice were measured by LC-MS/MS and compared to a standard curve of compound diluted into mouse plasma.

For PD analysis, mouse tissues were harvested after euthanizing and immediately frozen at −80 °C. The tissues were then homogenized in Triton lysis buffer using a FastPrep-24 homogenizer (MP Biomedicals) and a steel bead (Qiagen # 69989). Homogenates were processed for Western blotting as above.

**Statistical methods**. GraphPad Prism (version 7 for Windows) was used for all statistical analysis. PD results upon compound treatment were analyzed via one-way ANOVA.

**Data availability**. All relevant data are available from the authors. PDB IDs for crystal structures are as follows: Rheb-GDP (5YXH), Rheb with **1** (6BSX), and Rheb with **NR1** (6BT0).

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

## Acknowledgements

We thank James Hsieh, Jianing Xu, and David Sabatini for providing mTOR constructs. We thank David Sabatini, Brendan Manning, and Byron DeLaBarre for helpful discussions. We also thank Shomit Sengupta, Jessica Howell, and David O'Neill for helpful comments and suggestions. We thank the following organizations for performing portions of this study: HD Biosciences Co. Ltd., Viva Biotech Ltd., Beryllium Discovery Corp., Schrödinger, LLC, Shanghai ChemPartner Ltd., IniXium, and Evotec AG.

## Author contributions

S.J.M., S.N., L.M. and G.P.V. prepared the manuscript. S.J.M., S.N., and E.S. designed the NMR, X-ray, and ITC experiments. S.N. and E.S. designed the compounds and analyzed the structure activity relationship. S.J.M., L.M., L.A.B. and S.A.K. developed and/or performed biochemical and cell-based in vitro assays. S.N., S.A.K. and E.S. designed the in vivo experiments. S.J.M. and L.A.B. analyzed the tissue samples. G.P.V. and E.S. supervised this study.

## Additional information

**Competing interests:** All authors are employees of Navitor Pharmaceuticals, Inc.

