## [Peer Review File · Nature Communications]

Reviewers' comments:

Reviewer #1 (Remarks to the Author):

The manuscript „A Novel Small Molecule Inhibitor of Rheb Selectively Inhibits mTORC1 Signaling“ by Mahoney et al describes the first reported direct Rheb inhibitors and their characterization in biophysical, biochemical, cellular, and in vivo experiments. The disclosed Rheb inhibitor, NR1, is shown to bind to Rheb with micromolar affinity, inhibits only mTORC1 but not mTORC2 signaling, and shows PD effects in vivo.

This is an excellent paper which describes exciting data on a relevant system. I am impressed by the biophysical and biochemical validation of the inhibitor and the completeness of the data package. I have no doubts that the research is solid and experiments and conclusions are valid. Limitations of the manuscript are 1) Based on its properties, NR1 seems to be a tool compound only, not a development compound. 2) Only PD effects are shown in vivo, rather than efficacy models. 3) The activity of NR1 in cells is only double-digit micromolar (this should be pointed out in the manuscript).

Two other remarks: 1) Like Ras, Rheb is farnesylated and is natively embedded in the lysosomal membrane. Recent work on Ras has shown that farnesylation can have a dramatic impact on compound activity for certain mechanisms of action. I understand that the present work is carried out using truncated, non membrane-bound Rheb. The authors may add a short comment on this in the discussion. 2) The X-ray electron density of the bound ligands 1 and NR1 should be shown in the Supplementary Material.

The recently reported ligands for Ras, Rho and Ral have shown that small GTPases have lost their nimbus as undruggable targets. Therefore, the disclosure of a Rheb inhibitor may not have the impact as it would have had 5 years ago. Still, I recommend publication of this manuscript in Nature Communications because of the complete data package, the remarkable affinity of the compounds, and the novelty of drugging Rheb.

Reviewer #2 (Remarks to the Author):

The manuscript by Mahoney et al describes the identification of a chemical probe that interferes with RHEB-induced activation of mTORC1. The authors used FBLG and STD-NMR to screen for RHEB-binding compounds, and identified a weak binder (1) of the switch 2 region of RHEB which was further optimised to generate the more potent NR1. Their assay cascade also involved the development of a high throughput Rheb-coupled in vitro kinase assay.

NR1 is shown to interfere with Rheb-dependent activation of mTOR in cells, and to inhibit phosphorylation of the ribosomal S6 protein following fasting-refeeding in mice. Importantly, the authors show that treatment of prostate cancer PC3 cells with NR1 did not inhibit phosphorylation of Akt at the mTORC2 site (S473).

Inhibition of Rheb as a more selective approach to inhibition of mTORC1 is an attractive therapeutic avenue for the treatment of mTOR-associated pathologies. The importance of this selectivity is highlighted by the potential side effects of chronic exposure to rapalogues or mTOR kinase inhibitors.

Additionally, given the difficulties in identifying suitable binding pockets in Ras-family GTPases for the development of small molecule modulators, the identification of compounds that can directly target this family of proteins represents a significant advancement.

Although this manuscript describes a clever approach for the identification of Rheb inhibitors (the importance of which has already been mentioned) and delivers a compound with encouraging properties, some of the conclusions about the selectivity of the compound are based on minimal experimental evidence which can be explained by alternative hypotheses. Additionally, while targeting of Rheb could, in theory, phenocopy the therapeutic benefit of existing mTOR inhibitors while minimising the risk of side effects associated with chronic inhibition of mTORC2, targeting

Rheb could have its own problems. Although these concerns are beyond the scope of this study, the manuscript would benefit from a discussion of the potential effects of chronic Rheb inhibition (e.g. neuronal regeneration and the adaptive immune response). Finally, the mechanism of action of NR1 does not seem entirely clear from the data presented.

Below is a list of concerns that would need to be addressed in order for this work to be considered for publication in Nature Communications.

1. The authors state that it is "conceivable that portions of NR1 positioned outside the Rheb binding pocket sterically block the interaction with mTORC1. Given that the authors have access to recombinant Rheb and mTOR protein, it is unclear why there is no direct evidence that NR1 can inhibit their interaction. Alternatively, the effects of NR1 on Rheb-mTOR interactions could be shown in cells through co-IP. This is an important experiment in trying to understand the mechanism of action because Rheb could also be breaking the Rheb/Raptor interaction (PMID: 16631613).

2. The evidence for the selectivity of the compound is minimal. In fact, the only experiment that explores the selectivity of NR1 is a cell-based assay using MCF7 cells in which the inhibitor induces ERK1/2 phosphorylation. The authors argue that this provides evidence of NR1 not interfering with RAS function. However, RAS interacts with RalGDS through its switch 2 region and therefore Ral activation might be a better control. In the absence of direct measurements of NR1 binding to Rheb related GTPases (e.g. RAS, RALA/B, RASL, RAP) a better surrogate might be necessary to build confidence. Also, it is unclear whether NR1 is proposed to be selective for Rheb regulation of mTORC1. Although little is known about targets of Rheb other than mTOR, one protein shown to interact with and Rheb and whose stability seems to depend on this interaction is BACE1. The effects of NR1 on BACE1 should be interrogated in cells. On the issue of selectivity, it may also be important to evaluate the effects of NR1 on S6K activation in Rheb-deficient cells or in cells that express Rheb-independent mTOR mutants.

3. The data in figure 4D are the only evidence that NR1 does not interfere with mTORC2. Given that the kinetics of NR1-induced Rheb inhibition in cells was not shown, and because a single time point of drug treatment was examined, it is difficult to determine with certainty whether phosphorylation of Akt on Ser473 is unaffected by NR1. Also, PKC and/or SGK1 phosphorylation should be looked at as additional readouts of mTORC2 activity.

Other minor points to address

- a. Need justification for using MCF7 cells in some experiments and Jurkat cells in others.
- b. IC₅₀ for NR1 in the Rheb-IVK assay is 2.1 μ M and in cells the IC₅₀ for inhibition of S6K phosphorylation is 2 μ M. Some comments about the cell permeability of NR1 might be useful to contextualise this result.
- c. In Figure 4B (right panel), a pS6K control is necessary to document the effects of NR1 on EGF-induced mTORC1 activation.
- d. In figure 5A, rapamycin seems to have a more significantly pronounced effect on cell size compared to the highest dose of NR1 tested. Experiments in Figure 4 suggest that this dose of NR1 (10 μ M) was equipotent to 100nM rapamycin for S6K inhibition, and also for inhibition of protein synthesis according to Figure 5B. An explanation for the discrepancy in the biological effect (i.e. cell size) is needed.
- e. In figure 6, assessment of S6K and/or 4E-BP1 should also be included for confidence.

For the reasons stated above, publication of this manuscript in its current form is not recommended and should only be reconsidered if all issues listed have been properly addressed.

A Novel Small Molecule Inhibitor of Rheb Selectively Inhibits mTORC1 Signaling

Response to Reviewers' Comments

The authors wish to thank both reviewers for their careful consideration of the manuscript and their insightful comments. We have attempted to address all of the comments, which included significant new experimental data. Below, we detail our point-by-point responses to the comments.

Reviewer #1 (Remarks to the Author):

The manuscript “A Novel Small Molecule Inhibitor of Rheb Selectively Inhibits mTORC1 Signaling” by Mahoney et al describes the first reported direct Rheb inhibitors and their characterization in biophysical, biochemical, cellular, and in vivo experiments. The disclosed Rheb inhibitor, NR1, is shown to bind to Rheb with micromolar affinity, inhibits only mTORC1 but not mTORC2 signaling, and shows PD effects in vivo.

This is an excellent paper which describes exciting data on a relevant system. I am impressed by the biophysical and biochemical validation of the inhibitor and the completeness of the data package. I have no doubts that the research is solid and experiments and conclusions are valid.

Limitations of the manuscript are 1) Based on its properties, NR1 seems to be a tool compound only, not a development compound. 2) Only PD effects are shown in vivo, rather than efficacy models. 3) *The activity of NR1 in cells is only double-digit micromolar (this should be pointed out in the manuscript).*

Response: The reviewer is correct that we have shown only pharmacodynamics effects of NR1 since fully evaluating the compound in pharmacological efficacy models is outside the scope of this work. The potency of NR1 in the *in vitro* kinase and cellular assays is 2.1 and 2 μ M respectively; (i.e. single-digit micromolar) and this was mentioned in the Discussion section (p. 24) per the suggestion of Reviewer 1.

“Several lines of evidence suggest that **NR1** selectively engages Rheb in cells in the single-digit micromolar potency range.”

Two other remarks: 1) Like Ras, Rheb is farnesylated and is natively embedded in the lysosomal membrane. Recent work on Ras has shown that farnesylation can have a dramatic impact on compound activity for certain mechanisms of action. *I understand that the present work is carried out using truncated, non membrane-bound Rheb. The authors may add a short comment on this in the discussion.*

Response: Some studies in this work (X-ray crystallography, NMR and ITC) were carried out using truncated Rheb, while the *in vitro* kinase assay used the full-length protein. We also note that the C-terminal truncation does not impact the overall structure and conformation of Rheb as described (Ismail, S. *et al.* Arl2-GTP and Arl3-GTP regulate a GDI-like transport system for farnesylated cargo. *Nat Chem Biol* **7**, 942–949 (2011)).

2) *The X-ray electron density of the bound ligands 1 and NR1 should be shown in the Supplementary Material.*

Response: We have included the electron density plots for ligands 1 and NR1 (Supplementary Figures S2 and S7).

The recently reported ligands for Ras, Rho and Ral have shown that small GTPases have lost their nimbus as undruggable targets. Therefore, the disclosure of a Rheb inhibitor may not have the impact as it would have had 5 years ago. Still, I recommend publication of this manuscript in Nature Communications because of the complete data package, the remarkable affinity of the compounds, and the novelty of drugging Rheb.

Reviewer #2 (Remarks to the Author):

The manuscript by Mahoney et al describes the identification of a chemical probe that interferes with RHEB-induced activation of mTORC1. The authors used FBLG and STD-NMR to screen for RHEB-binding compounds, and identified a weak binder (1) of the switch 2 region of RHEB which was further optimised to generate the more potent NR1. Their assay cascade also involved the development of a high throughput Rheb-coupled in vitro kinase assay.

NR1 is shown to interfere with Rheb-dependent activation of mTOR in cells, and to inhibit phosphorylation of the ribosomal S6 protein following fasting-refeeding in mice. Importantly, the authors show that treatment of prostate cancer PC3 cells with NR1 did not inhibit phosphorylation of Akt at the mTORC2 site (S473).

Inhibition of Rheb as a more selective approach to inhibition of mTORC1 is an attractive therapeutic avenue for the treatment of mTOR-associated pathologies. The importance of this selectivity is highlighted by the potential side effects of chronic exposure to rapalogues or mTOR kinase inhibitors.

Additionally, given the difficulties in identifying suitable binding pockets in Ras-family GTPases for the development of small molecule modulators, the identification of compounds that can directly target this family of proteins represents a significant advancement.

Although this manuscript describes a clever approach for the identification of Rheb inhibitors (the importance of which has already been mentioned) and delivers a compound with encouraging properties, some of the conclusions about the selectivity of the compound are based on minimal experimental evidence which can be explained by alternative hypotheses. Additionally, while targeting of Rheb could, in theory, phenocopy the therapeutic benefit of existing mTOR inhibitors while minimising the risk of side effects associated with chronic inhibition of mTORC2, targeting Rheb could have its own problems. *Although these concerns are beyond the scope of this study, the manuscript would benefit from a discussion of the potential effects of chronic Rheb inhibition (e.g. neuronal regeneration and the adaptive immune response).*

Response: We agree with Reviewer 2 and have included a brief discussion on the potential effects of chronic Rheb inhibition in the discussion section (p. 25).

“As with any new pharmacological agent, the anticipated side effects will need to be considered. The only data available from the literature that may help in understanding the potential side-effects of reducing Rheb activity come from genetic knock-out studies which, because of their inherent impacts on developmental and other effects, cannot be directly related to what might be expected following pharmacological inhibition. Thus, careful development of a novel Rheb inhibitor would have to take into account the potential organ systems that may be impacted, such as neurological^{63, 64} and immunological^{65, 66}.”

Finally, the mechanism of action of NR1 does not seem entirely clear from the data presented.

Below is a list of concerns that would need to be addressed in order for this work to be considered for publication in Nature Communications.

1. *The authors state that it is “conceivable that portions of NR1 positioned outside the Rheb binding pocket sterically block the interaction with mTORC1. Given that the authors have access to recombinant Rheb and mTOR protein, it is unclear why there is no direct evidence that NR1 can inhibit their interaction. Alternatively, the effects of NR1 on Rheb-mTOR interactions could be shown in cells through co-IP. This is an important experiment in trying to understand the mechanism of action because Rheb could also be breaking the Rheb/Raptor interaction (PMID: 16631613).*

Response: We agree with Reviewer #2 that there is no direct evidence that **NR1** inhibits the interaction between Rheb and mTORC1. As suggested by the Reviewer, we attempted the co-immunoprecipitation endogenous mTOR and Rheb but were unable to detect the native complex in cell lysates. The inability to detect a complex of endogenous proteins is consistent with the lack of published data to this effect and verbal communications from several laboratories studying the mTOR pathway, including the Manning and Sabatini labs. We further attempted the same experiment using overexpressed FLAG-Rheb and myc-mTOR, keeping in mind that this situation may not be physiologically relevant because the other components of the mTORC1 complex are not present. The interaction could be detected in this case. However, **NR1** does not modulate it (**Figure 1, review-only material**). Based on these data, we are unable to conclude whether NR1 blocks the Rheb-mTOR interaction in a physiologically relevant setting. Accordingly, we have removed statements to this effect from the abstract and other sections of the manuscript. *We have not investigated the possibility that NR1 blocks the Rheb-Raptor interaction.*

2. The evidence for the selectivity of the compound is minimal. In fact, the only experiment that explores the selectivity of NR1 is a cell-based assay using MCF7 cells in which the inhibitor induces ERK1/2 phosphorylation. The authors argue that this provides evidence of NR1 not interfering with RAS function. However, RAS interacts with RalGDS through its switch 2 region

and therefore Ral activation might be a better control. *In the absence of direct measurements of NR1 binding to Rheb related GTPases (e.g. RAS, RALA/B, RASL, RAP) a better surrogate might be necessary to build confidence.*

Response: We agree that additional evidence for the selectivity of **NR1** against other GTPases would add substantially to the manuscript. Accordingly, we have evaluated the effect of NR1 on the activity of Rap1. **NR1** has no impact on the activity of Rap1 in an *in vitro* Rap1 pulldown assay as shown in **Figure 4C**).

Also, it is unclear whether NR1 is proposed to be selective for Rheb regulation of mTORC1. Although little is known about targets of Rheb other than mTOR, one protein shown to interact with and Rheb and whose stability seems to depend on this interaction is BACE1. *The effects of NR1 on BACE1 should be interrogated in cells.*

Response: The current work is focused on inhibition of the mTOR pathway by a Rheb inhibitor. Therefore, we have not investigated the effect of **NR1** on targets other than mTOR such as BACE1. Although a potentially interesting experiment, we respectfully believe that this is outside the scope of the present work.

On the issue of selectivity, it may also be important to evaluate the effects of NR1 on S6K activation in Rheb-deficient cells or in cells that express Rheb-independent mTOR mutants.

Response: We agree that this is an important point and have evaluated the effect of **NR1** on mTORC1 activation in transiently transfected Rheb-independent mTOR mutant cells as previously described (Xu, *et al.*, Mechanistically distinct cancer-associated mTOR activation clusters predict sensitivity to rapamycin. *J Clin Invest.* 2016;126(9):3526-40). **NR1** has no impact on pS6K1 in Rheb-independent mutants, which strongly supports the proposed mechanism of action. This new data has been added to the manuscript (**Figure 4E**).

3. *The data in figure 4D are the only evidence that NR1 does not interfere with mTORC2. Given that the kinetics of NR1-induced Rheb inhibition in cells was not shown, and because a single time point of drug treatment was examined, it is difficult to determine with certainty whether phosphorylation of Akt on Ser473 is unaffected by NR1.*

Response: As suggested by Reviewer #2, we have performed a full time-course experiment on the effect of **NR1** on pAkt in PC3 cells. **NR1** inhibits pS6K1 and activates pAkt at time points ranging from 2 h through 24 h, which is consistent with the effect of rapamycin on pAkt over the same time course. These new data have been added to the manuscript (**Figure 4F**).

Also, PKC and/or SGK1 phosphorylation should be looked at as additional readouts of mTORC2 activity.

Response: We further evaluated the effect of **NR1**, long-term rapamycin, and Torin-1 on pSGK1 and pPKC ζ in PC3 cells. The pPKC ζ antibody we used had been used in a previous publication regarding mTORC2 phosphorylation of PKC ζ (Li, *et al.* mTORC2

phosphorylates protein kinase C ζ to regulate its stability and activity. *EMBO Rep.* 2014;2:191-8). Unfortunately, this experiment was inconclusive because the antibodies were not detecting bands that were reduced using Torin-1, a potent mTORC2 inhibitor (or long-term treatment with rapamycin in PC3 cells). This may be due to improper specificity of the antibodies that were available to us or an incomplete understanding of the nature of the phosphorylation event. Nevertheless, we have included this data for review (**Figure 2, review only material**).

Other minor points to address

a. Need justification for using MCF7 cells in some experiments and Jurkat cells in others.

Response: We have added a comment addressing this question in the results section regarding the cell size assay (p. 18).

“Based on the dependence of cell size on mTORC1 activity⁴⁹, we assessed the effect of **NR1** on the size of Jurkat cells, a human T cell line grown in suspension and responsive to mTORC1 inhibition.”

b. IC50 for NR1 in the Rheb-IVK assay is 2.1 μ M and in cells the IC50 for inhibition of S6K phosphorylation is 2 μ M. *Some comments about the cell permeability of NR1 might be useful to contextualise this result.*

Response: As Reviewer #2 rightly points out, the biochemical and cell-based potencies of **NR1** are in the same range, suggesting that high cell permeability may be playing a role. We have added a comment to this effect in the results section regarding **NR1** activity in cells (p. 15).

“It is notable that the cellular potency of **NR1** is in the same range as that of its biochemical activity in the Rheb-IVK strongly indicating a high degree of cell permeability of the compound.”

c. *In Figure 4B (right panel), a pS6K control is necessary to document the effects of NR1 on EGF-induced mTORC1 activation.*

Response: In the EGF-induced mTORC1 activation experiment, we have included a pS6K control along with pERK. The original figure in the manuscript has been replaced with a new figure with the additional data (**Fig. 4B**).

d. In figure 5A, rapamycin seems to have a more significantly pronounced effect on cell size compared to the highest dose of NR1 tested. Experiments in Figure 4 suggest that this dose of NR1 (10 μ M) was equipotent to 100nM rapamycin for S6K inhibition, and also for inhibition of protein synthesis according to Figure 5B. *An explanation for the discrepancy in the biological effect (i.e. cell size) is needed.*

Response: **Figure 4A** now includes a darker exposure of pS6K1 to indicate that at 10 μ M, **NR1** does not inhibit S6K1 phosphorylation as well as 100 nM rapamycin. This difference in activity level could account for the less robust effect of 10 μ M **NR1** in the cell size assay

in **Figure 5A**. it should be noted that there was no expectation that the inhibition of Rheb function would be equivalent to the inhibition of the mTORC1 complex by rapamycin.

e. In figure 6, assessment of S6K and/or 4E-BP1 should also be included for confidence.

Response: As recommended by reviewer #2, we have measured the effect of **NR1** treatment on 4E-BP1 in mouse gastrocnemius muscle. Consistent with pS6K1, clear modulation of 4E-BP1 is observed, which adds further confidence to the *in vivo* efficacy of **NR1**. For the kidney, our 4E-BP1 and p4E-BP1 antibodies did not work as well. As such, we have moved the kidney data to the supplementary data (**Supplementary Fig. S10**).

REVIEWERS' COMMENTS:

Reviewer #2 (Remarks to the Author):

The authors have appropriately addressed all the previously raised issues and I am now happy to recommend publication of the revised manuscript.